# Pharmacokinetics of Levodopa and 3-O-Methyldopa in Parkinsonian Patients Treated with Levodopa and Ropinirole and in Patients with Motor Complications

**DOI:** 10.3390/pharmaceutics13091395

**Published:** 2021-09-03

**Authors:** Urszula Adamiak-Giera, Wojciech Jawień, Anna Pierzchlińska, Monika Białecka, Jan Dariusz Kobierski, Tomasz Janus, Barbara Gawrońska-Szklarz

**Affiliations:** 1Department of Pharmacokinetics and Therapeutic Drug Monitoring, Pomeranian Medical University, al. Powstańców Wlkp. 72, 70-111 Szczecin, Poland; pierzchlinska@gmail.com (A.P.); monika-bialecka@post.pl (M.B.); 2Department of Pharmaceutical Biophysics, Jagiellonian University Medical College, ul. Medyczna 9, 30-688 Kraków, Poland; wojciech.jawien@uj.edu.pl (W.J.); jan.kobiersk@uj.edu.pl (J.D.K.); 3Department of Clinical Toxicology, Pomeranian Medical University, al. Powstańców Wlkp. 72, 70-111 Szczecin, Poland; tjanus@pum.edu.pl

**Keywords:** Parkinson’s disease, pharmacokinetics, levodopa, 3-O-methyldopa, ropinirole, motor complications

## Abstract

Parkinson’s disease (PD) is a progressive, neurodegenerative disorder primarily affecting dopaminergic neuronal systems, with impaired motor function as a consequence. The most effective treatment for PD remains the administration of oral levodopa (LD). Long-term LD treatment is frequently associated with motor fluctuations and dyskinesias, which exert a serious impact on a patient’s quality of life. The aim of our study was to determine the pharmacokinetics of LD: used as monotherapy or in combination with ropinirole, in patients with advanced PD. Furthermore, an effect of ropinirole on the pharmacokinetics of 3-OMD (a major LD metabolite) was assessed. We also investigated the correlation between the pharmacokinetic parameters of LD and 3-OMD and the occurrence of motor complications. Twenty-seven patients with idiopathic PD participated in the study. Thirteen patients received both LD and ropinirole, and fourteen administered LD monotherapy. Among 27 patients, twelve experienced fluctuations and/or dyskinesias, whereas fifteen were free of motor complications. Inter- and intra-individual variation in the LD and 3-OMD concentrations were observed. There were no significant differences in the LD and 3-OMD concentrations between the patients treated with a combined therapy of LD and ropinirole, and LD monotherapy. There were no significant differences in the LD concentrations in patients with and without motor complications; however, plasma 3-OMD levels were significantly higher in patients with motor complications. A linear one-compartment pharmacokinetic model with the first-order absorption was adopted for LD and 3-OMD. Only mean exit (residence) time for 3-OMD was significantly shorter in patients treated with ropinirole. Lag time, *V/F*, *CL/F* and *t*_max_ of LD had significantly lower values in patients with motor complications. On the other hand, *AUC* were significantly higher in these patients, both for LD and 3-OMD. 3-OMD *C*_max_ was significantly higher in patients with motor complications as well. Our results showed that ropinirole does not influence LD or 3-OMD concentrations. Higher 3-OMD levels play a role in inducing motor complications during long-term levodopa therapy.

## 1. Introduction

Parkinson’s disease (PD) is a common neurological disorder affecting 2–3% of the population 65 years of age and older [1]. The disease is characterized by progressive degeneration of the dopaminergic nigrostriatal system and depletion of dopamine, which results in the core motor symptoms of bradykinesia, rigidity, tremor, and postural instability [2,3,4]. Pharmacotherapy of PD constitutes dopaminergic drugs, mainly the dopamine precursor levodopa and dopamine receptor agonists. Levodopa (L-dopa, LD) is one of the effective agents in the management of Parkinson’s disease and reaches its site of action by crossing the blood–brain barrier (BBB) [5,6,7]. In brain tissues, LD is decarboxylated to dopamine, which is typically stored in presynaptic terminals of striatal neurons [5,7,8]. LD is primarily metabolized by peripheral amino acid decarboxylase (AADC), monoamine oxidase (MAO), and catechol O-methyltransferase (COMT) [6]. Although the inhibition of AADC is a common clinical strategy during LD therapy, it has been demonstrated that only 5–10% of an oral dose of LD reaches the central nervous system (CNS) when combined with an AADC inhibitor [9,10]. The co-administration of AADC inhibitors, such as benserazide and carbidopa, prolongs the efficacy and promotes the tolerability of LD [1,11,12]. A major metabolite of levodopa is 3-O-methyldopa (3-OMD), formed by COMT. 3-OMD is a competitive inhibitor of LD intestinal absorption, BBB transport, and is a competitive substrate for CNS dopamine uptake [2,5,8,13,14]. Pharmacokinetic (PK) studies have shown a long plasma half-life for 3-OMD, therefore, it accumulates in particular during repeated LD administration and may compete with levodopa at the large neutral amino acid transport carriers of the gastrointestinal tract and of the blood–brain barrier. Therefore elevated 3-OMD levels may contribute to a reduced levodopa delivery to the blood and to the brain [2,8,14].

Long-term levodopa treatment of PD is frequently associated with motor fluctuations and dyskinesias, causing a serious impact on a patient’s quality of life [1,2,15,16]. One of the possible means to delay motor complications is to include other classes of antiparkinsonian drugs such as dopamine agonists (DA), catechol-O-methyltransferase (COMT) inhibitors, or monoamine oxidase type B (MAO-B) inhibitors into the treatment regime [1,8]. Ropinirole is a non-ergoline dopamine agonist developed as a therapeutic agent for the treatment of the signs and symptoms of Parkinson’s disease [8,17]. It has been shown that the addition of ropinirole allows improving a patient’s clinical condition without increasing the LD dosage [18]. It seems plausible that ropinirole may influence 3-OMD concentration, being largely responsible for the occurrence of motor complications. Risk and time to the emergence of motor complications vary substantially among patients for complex reasons, including both disease and drug-related factors, particularly treatment with levodopa [2,19,20]. A better understanding of the LD plasma concentration-effect relationship could be valuable in the assessment of PD management [4,8,21,22]. In the advanced stages of Parkinson’s disease, fluctuations in response to LD reflect the changes in LD plasma levels [2,4,8,9,21,22,23].

The aim of our study was to determine the pharmacokinetics of LD: used in monotherapy or in combination with ropinirole, in patients with advanced PD. Furthermore, an effect of ropinirole on the pharmacokinetics of 3-OMD (a major LD metabolite) was assessed. We also investigated the correlation between the pharmacokinetic parameters of levodopa and 3-OMD and the occurrence of motor complications.

## 2. Materials and Methods

### 2.1. Patients

Twenty-seven (20 men and 7 women) patients aged from 45 to 86 (69.7 ± 9.2), bodyweight from 50 to 113 (79.6 ± 18.9), with idiopathic PD participated in the study. The protocol was approved by the ethics committee (Pomeranian Medical University in Szczecin, Szczecin, Poland) and all patients gave written consent before any procedures were performed.

The subjects were diagnosed with idiopathic PD according to UK Parkinson’s disease Society Brain Bank clinical diagnostic criteria [24]. All patients with clinical symptoms suggesting secondary causes of the parkinsonian syndrome (vascular, drug-induced), with features suggestive of atypical parkinsonian syndromes (multiple system atrophy, progressive supranuclear palsy, and corticobasal syndrome) or with the presence of cardiovascular disease (e.g., stroke, heart failure) were excluded from final data analysis. Informed written consent was obtained before participation.

Demographic and clinical data were collected according to a semi-structured interview and medical documentation. Based on the received antiparkinsonian treatment, subjects were divided into two groups: administrating both levodopa and ropinirole (*n* = 13), and administrating levodopa but not ropinirole (*n* = 14). The characteristics of both groups were compared in Table 1. Independently, all the subjects were divided into two other groups, based on the presence of motor complications: dyskinesias and/or fluctuations and absence of these complications (Table 2).

### 2.2. Study Design

All current antiparkinsonian medication (levodopa, ropinirole) was withdrawn 12 h before the study began. After an overnight fast, in the morning of the study day, patients received an indwelling venous cannula in the antecubital vein. After a baseline investigation of motor status and withdrawal of blood sample, levodopa or levodopa and ropinirole were given orally with a glass (250 mL) of water. The patients received a single dose containing 100 mg of levodopa plus 25 mg of benserazide or 25 mg of carbidopa, the same medication as during their stable treatment. Although for some of the patients the usual morning LD dosage was higher, in the pharmacokinetic studies, it is advised to use 100 mg of LD. Subjects who had ropinirole in their treatment regime received 24 h prolonged release in the same dose, from 4 mg to 10 mg. Blood samples were taken before and 30, 45, 60, 90, and 120 min after dosing. Ropinirole was measured in the same samples before treatment and 60, 120 min after dosing. One hour after the administration of levodopa or levodopa and ropinirole, patients received a standardized breakfast (one sandwich and 250 mL of water).

#### Determination of Plasma Concentrations of Levodopa, 3-OMD, and Ropinirole

Blood samples for determination of concentrations of levodopa, 3-OMD, and ropinirole in plasma were collected into 4 mL tubes containing EDTA and were immediately chilled. Samples were separated by centrifugation at 4 °C (1500× *g* for 10 min). Plasma was transferred into polypropylene tubes and immediately frozen at −30 °C until analysis within 14 days. Levodopa and 3-OMD plasma concentrations were determined by specific reversed-phase high-performance liquid chromatographic method with electrochemical detection, according to the technique previously reported by Saxer [25] with our own modification [26]. Chromatographic separation was carried out on analytical column C18 ThermoHypersil Gold 3 mm (3.0 × 100 mm) (Thermo). The limit of quantitation amounted to 0.025 mg/L for levodopa and 0.1 mg/L for 3-OMD. A rapid and sensitive liquid chromatography-mass spectrometry method was applied to the determination of ropinirole in plasma samples according to a previously published method [27].

### 2.3. Mathematical Modelling and Parameter Estimation of Levodopa and 3-OMD

#### 2.3.1. Pharmacokinetic Model for the Drug-Metabolite System

A linear one-compartment pharmacokinetic model with the first-order absorption was adopted for the parent drug (Figure 1). The presence of the lag-time, $T_{lag}$ was assumed. The second compartment was postulated for the main metabolite (3-OMD). It is assumed that these compartments correspond to the same physical space, therefore, both compartments have the same volume, *V*.

The remaining symbols have the following meaning: $k_{10}$—first-order elimination constant of levodopa on all routes other than the metabolism to 3-OMD, $k_{12}$—first-order constant of the levodopa metabolism to 3-OMD, $k_{20}$—first-order elimination constant of 3-OMD, $k_{a}$—first-order absorption constant, *F*—fraction of the dose that is absorbed, *D*—an administered dose of levodopa. Differential equations of the model, as well as the model definition expressed in the MlxTran language [28], are given in the Appendix A. There were also attempts to apply other models, in particular, the two-compartment model for 3-OMD was investigated. These attempts, however, proved to be unsuccessful. Steady-state conditions were postulated, i.e., it was assumed that the patients regularly followed the prescribed dosing regimen for at least 10 days before the study.

#### 2.3.2. Population PK Modelling

With the aid of the Monolix [29] software release 2019R1 (Lixoft^®^, Antony, France), the population pharmacokinetic model has been implemented for the drug-metabolite system. The pharmacokinetic parameters of the model were estimated and their dependence on covariates evaluated. The following continuous variables: age, height, weight, creatinine concentration, ropinirole concentration 2 h after its administration (if any) as well as categorized variables: sex, motor complications and ropinirole therapy were available to be considered as covariates. Taking advantage of the automatic model selection feature available in 2109R1software release, the optimal set of covariates and an optimal model of the residual error were determined.

Parameters of the population model as well as the individual PK parameters of the patients were estimated using stochastic approximation estimation method (SAEM [30,31]) implemented in the Monolix program [29]. Standard deviations of the estimates and correlations between them were estimated based on the Fisher information matrix (FIM). The linearization procedure [32] appeared necessary in order to estimate the FIM. The Wald test was used to verify the significance of the covariates. Based on primary individual PK parameters, i.e., parameters of the PK model (see Figure 1), secondary parameters with more direct clinical meaning were calculated. These included the following parameters of levodopa: clearance (Cl/F), $AUC_{0-\infty }$, maximum concentration ($C_{\max}$) predicted after the single standard dose along with time ($t_{\max}$) after which it is attained, and mean residence time (MRT). The analogous parameters for 3-OMD were: $AUC_{3-OMD},\;t_{\max\;3-\mathrm{OMD}}$ and $C_{\max\;3-\mathrm{OMD}}$. In addition, the mean exit time (MET), the parameter which replaces MRT in the case of metabolite, was also calculated. The AUC, $C_{\max}$ and $t_{\max}$ parameters of both compounds are administration schedule dependent. Because the patients had very diverse dosing schemas, the presentation of these parameters values in the steady-state could not be standardized. Therefore, the parameters for the single-dose administration were calculated instead. The equations used for the calculation of the secondary parameters are given in the Appendix A. Internal model evaluation [33] included observations vs. predictions plots, NPDE analysis [34], NPDE normality plots, and visual predictive checks [29].

### 2.4. Statistical Analysis

Demographic parameters were described statistically as mean ± standard deviation (SD) values. Mann–Whitney test was used for comparisons of majority demographic characteristics between the analyzed groups of patients. The exact Fisher test was used for comparisons of binary data: sex and motor complications. Concentrations of levodopa and 3-OMD were described statistically as mean (SD) values.

For those PK parameters that were not part of the population model (secondary parameters), analysis of variance (ANOVA) with log-transformed data was used to detect significant effects. The log transformation is consistent with standard assumptions on PK parameters used in population modeling. All PK parameters were presented as mean ± SD. Statistical calculations were performed using the TIBCO Statistica 13.3 software package (TIBCO, Palo Alto, CA, USA) and SAS 9.4 system (SAS Institute, Cary, NC, USA).

## 3. Results

### 3.1. Patient Characteristics

The main characteristics of 13 patients who received both levodopa and ropinirole, and 14 patients who were not treated with ropinirole, are presented in Table 1. No significant differences were found between the analyzed parameters.

The main characteristics of 12 patients with motor complications (fluctuations and/or dyskinesias) and 15 patients without motor complications are presented in Table 2. Patients with motor complications showed a significantly longer duration of the disease (*p* < 0.001), and the current dose of LD was significantly higher (*p* < 0.001).

### 3.2. Plasma Concentration of Levodopa and 3-OMD

Mean plasma LD and 3-OMD concentrations in patients treated with LD and ropinirole (*n* = 13) and LD but not ropinirole (*n* = 14) are shown in Figure 2 and Figure 3. In addition, all measured concentrations are depicted in Appendix A in the Appendix A. Inter- and intra-individual variation in the LD and 3-OMD concentrations were observed. There were no significant differences in the LD concentrations between the analyzed groups. The highest concentration of LD was observed after 30 min in all patients treated with LD and ropinirole (Figure 2).

Plasma 3-OMD concentrations were similar in both groups and did not differ significantly (Figure 3).

Plasma LD and 3-OMD concentrations in patients with motor complications (*n* = 12) and without motor complications (*n* = 15) are shown in Figure 4 and Figure 5. There were no significant differences in the LD concentrations between the analyzed groups. The highest concentration of LD was observed after 30 min in patients with motor complications (Figure 4). Plasma 3-OMD concentrations were significantly higher in patients with motor complications over the entire range of observation (*p* < 0.05).

### 3.3. Pharmacokinetic Population Modelling

The optimal set of covariates, determined with the aid of automatic model selection by Monolix 2019R1 contained: sex and ropinirole concentration (2 h after the administration) as predictors for the parameter *V/F*, age for $k_{10}$, weight and movement disorder for $T_{lag}$, ropinirole therapy for $k_{20}$. The full form of the final population PK model equations and their parameter estimates are given in the Appendix A. Results of the Wald test are compiled in Table 3. They confirm the significance of all chosen covariates.

Results of the internal model evaluation are contained in the Appendix A.

### 3.4. Pharmacokinetic Parameters of Levodopa and 3-OMD

Pharmacokinetic parameters of levodopa and 3-OMD after administration levodopa and ropinirole and levodopa but not ropinirole are presented in Table 4. Pharmacokinetic parameters of levodopa and 3-OMD of patients with motor complications (fluctuations and/or dyskinesias) and without motor complications are presented in Table 5.

Only the mean exit time for 3-OMD (*MET*_3-OMD_) was significantly shorter in patients treated with ropinirole. The remaining parameters showed no statistically significant differences.

Lag time, *V/F*, *CL/F*, and *t*_max_ of LD proved to have significantly lower values in patients with motor complications. On the other hand, *AUC* was significantly higher in these patients, both for LD and 3-OMD. 3-OMD C_max_ was significantly higher only in patients with motor complications. For the remaining parameters, no statistically significant differences have been detected. It should be reminded that *AUC*_0–∞_, $t_{\max}$ and $C_{\max}$ were calculated for the case of a single administration of the standard dose, as described in the Section 2.

## 4. Discussion

Long-term PD treatment with dopaminergic medication poses a risk of developing motor complications. Several risk factors have been established, including individual factors (age, duration of the disease), and the daily LD dosage, the treatment duration, the medication being used.

We have established that patients with motor complications showed a significantly longer duration of the disease (*p* < 0.001), received significantly higher doses of LD, and the current dose of LD was significantly higher (*p* < 0.001), which is in concordance with other analyses [1,2,4].

The available literature suggests a correlation between the variability in LD pharmacokinetics and motor complications associated with the treatment [5,6,7,16]. One of the available methods to delay the motor complications is to combine the therapy with a dopamine agonist, e.g., ropinirole.

Explicit attention is focused on a major metabolite of LD—3-OMD. The role of 3-OMD in the pharmacokinetic–pharmacodynamic relationship of LD is still under investigation. It is not entirely clear how ropinirole affects the 3-OMD pharmacokinetics. We have presented that the concentration of LD and its metabolite 3-OMD were higher in patients treated with the combination therapy (levodopa and ropinirole) compared to the monotherapy (Figure 2 and Figure 3) but the differences were not statistically significant. The pharmacokinetic parameters of LD did not differ between the groups. Similarly, there were no significant differences in the pharmacokinetic parameters of 3-OMD, except for the mean exit time of 3-OMD, which was significantly shorter in the patients on the combined therapy. Slightly higher 3-OMD concentrations in the patients treated with ropinirole may indicate a predominance of the COMT-related LD metabolism. Thus, an inhibitory effect of ropinirole on the AADC activity cannot be excluded [35,36]. Indeed, Taylor et al. [37] showed that the administration of ropinirole in patients treated with LD without the AADC inhibitors increased the LD *C*_max_ by 16%. In our study, there was a nonsignificant increase in the LD C_max_ in the patients treated with ropinirole. We observed a significantly shorter mean-residence-time for 3-O-methyldopa in patients treated with LD and ropinirole, which may suggest a faster elimination. A detailed discussion of population model parameters is given in the Appendix A. Due to the small number of patients studied, we did not analyze the effect of ropinirole on motor complications. Population pharmacokinetics of ropinirole have demonstrated that sex, mild or moderate renal impairment, Parkinson’s disease stage, and concomitant illnesses, or the use of several common concomitant medications have no effect on the pharmacokinetics of ropinirole [37,38].

After the analysis of the LD pharmacokinetic parameters in patients with motor complications, we have determined significantly higher *AUC*, while lag time, *V/F*, *CL/F*, and *t*_max_ were significantly decreased. *C*_max_ LD was no significantly higher in these patients. Higher *C*_max_, a large *AUC*_0–∞_ (*p* < 0.01), a smaller *V/F*, and *CL/F* (*p* < 0.01) may be explained partly because the mean body weight of this group (68.2 ± 12.3 kg) was lower than that of patients without motor complications (88.7 ± 18.6 kg). High maximum LD concentration may lead to pulsatile stimulation of dopamine receptors in the CNS, a probable main cause of motor complications. The automatic model selection procedure selected neither age nor body weight as statistically significant covariates of the population model for V/F, as described in detail in the Appendix A. The patient’s age but not the weight proved to be one of predictors of k_10_ and, consequently, of the renal part of clearance (it was the total clearance that was reported by us), *C*_max_, and *AUC*.

We observed a very high accumulation of 3-OMD in patients treated with LD. As reported by other authors, our results showed that 3-OMD, which is a major metabolite of levodopa, plays a role in inducing motor complications during long-term LD therapy. We also reported a significant increase in *C*_max 3-OMD_ and *AUC*_3-OMD [0–∞]_. Lee et al. [13] tested the role of 3-OMD in the motor complications of levodopa therapy and showed that 3-OMD accumulation from long-term LD treatment may be involved in the adverse effects. Moreover, LD treatment might accelerate the progression of PD, at least in part, by 3-OMD. Our previous study suggested that diminished concentration of 3-OMD in plasma might contribute to an improvement of clinical state [39].

Some LD and 3-OMD pharmacokinetic parameters differed significantly in patients with motor complications compared to patients without them. Over 90% of PD patients are treated with oral LD. The pharmacokinetics of orally administered levodopa depends on gastric emptying because the absorption occurs only in the proximal one-third of the small intestine, not in the stomach [6,8]. The motility of the stomach varies with fed and fasted state, and erratic gastric emptying gives unpredictability to the LD concentration-time curve. Moreover, gastric emptying time is delayed in some PD patients compared to the control population, especially in patients with motor fluctuations [40,41]. In our research, significantly shorter lag time and time of *C*_max_ were observed in patients with motor complications. Due to many factors influencing the pharmacokinetic parameters of LD, each patient should be evaluated individually.

## 5. Conclusions

In the performed analyses, no significant effect of ropinirole on LD or 3-OMD concentrations have been established. However, an influence of ropinirole on 3-OMD metabolism cannot be excluded. A correlation between 3-OMD and motor complications was confirmed.

## Figures and Tables

**Figure 1 pharmaceutics-13-01395-f001:**
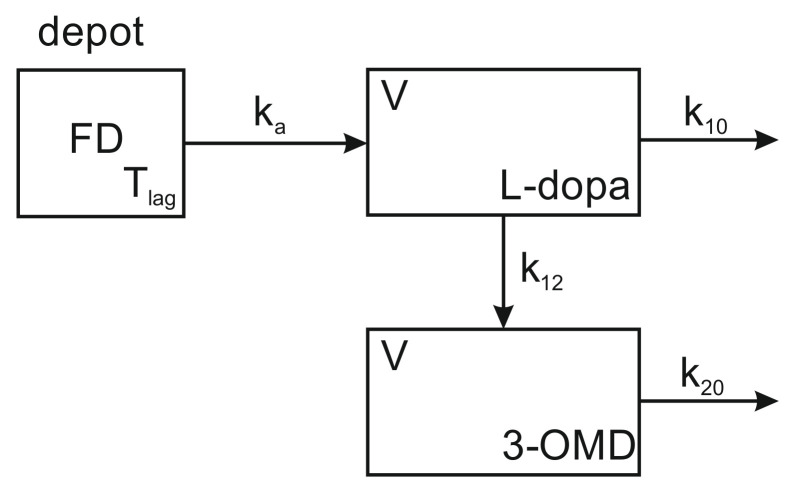
Pharmacokinetic model for the drug-metabolite system (L-dopa—3-OMD).

**Figure 2 pharmaceutics-13-01395-f002:**
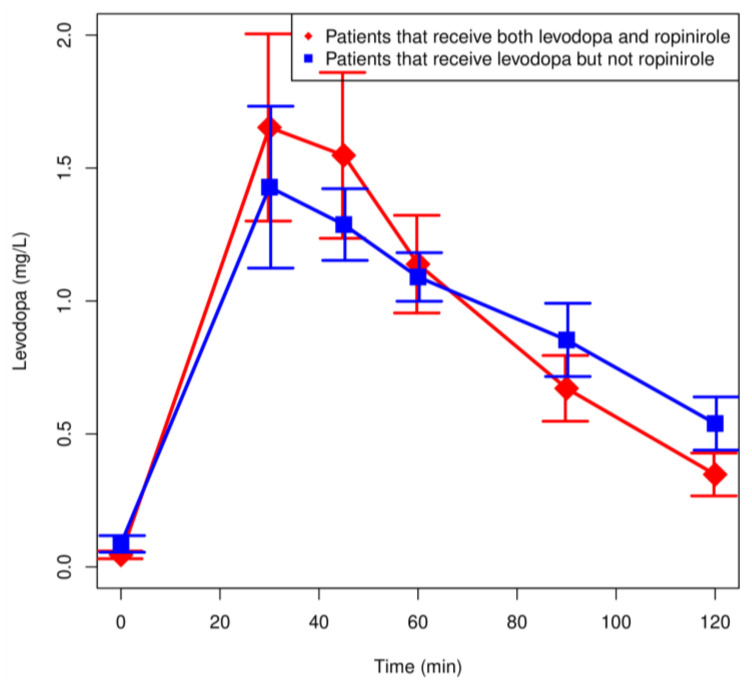
Plasma concentration: time profiles (mean values ± SEM) for levodopa of patients receiving both levodopa and ropinirole (*n* = 13) and patients receiving levodopa but no ropinirole (*n* = 14).

**Figure 3 pharmaceutics-13-01395-f003:**
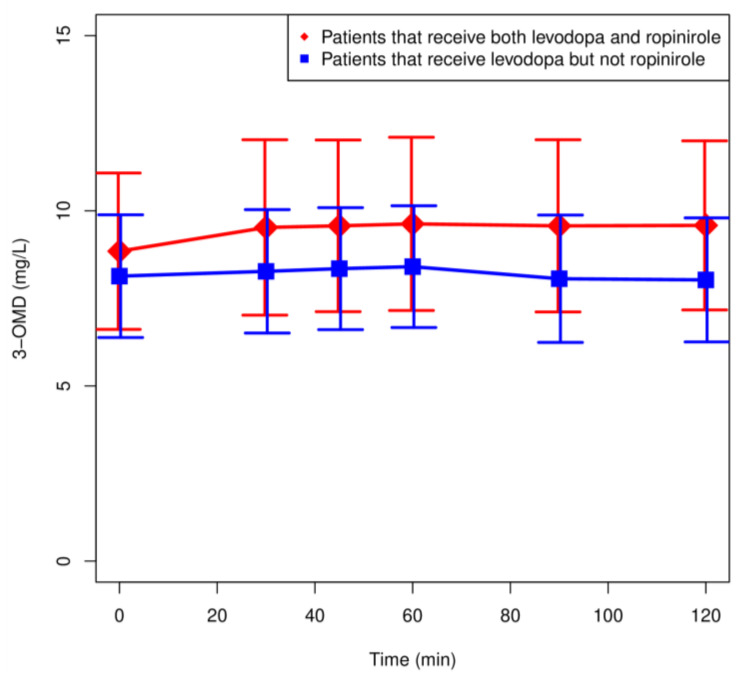
Plasma concentration: time profiles (mean values ± SEM) for 3-OMD of patients administrating both levodopa and ropinirole (*n* = 13) and patients administrating levodopa but no ropinirole (*n* = 14).

**Figure 4 pharmaceutics-13-01395-f004:**
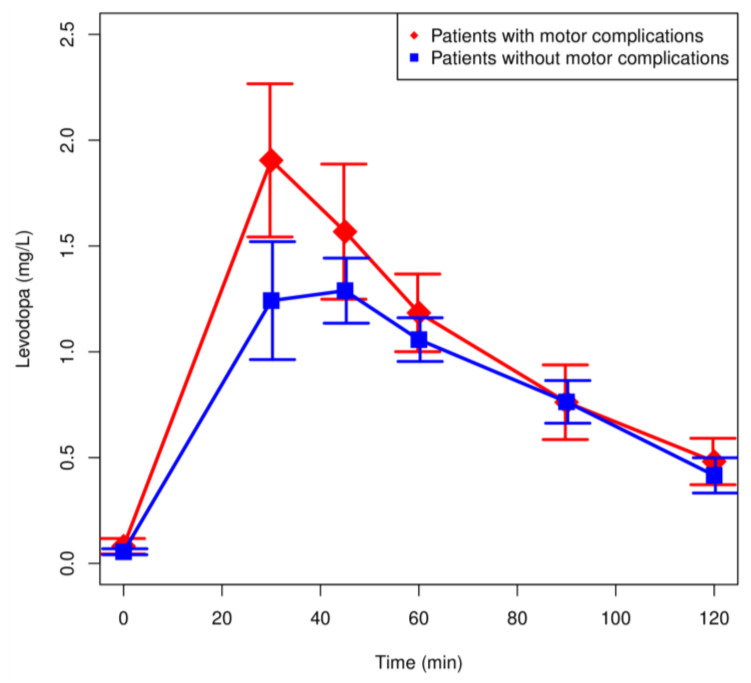
Plasma concentration: time profiles (mean values ± SEM) for levodopa of patients with motor complications (fluctuations and/or dyskinesias) and without motor complications.

**Figure 5 pharmaceutics-13-01395-f005:**
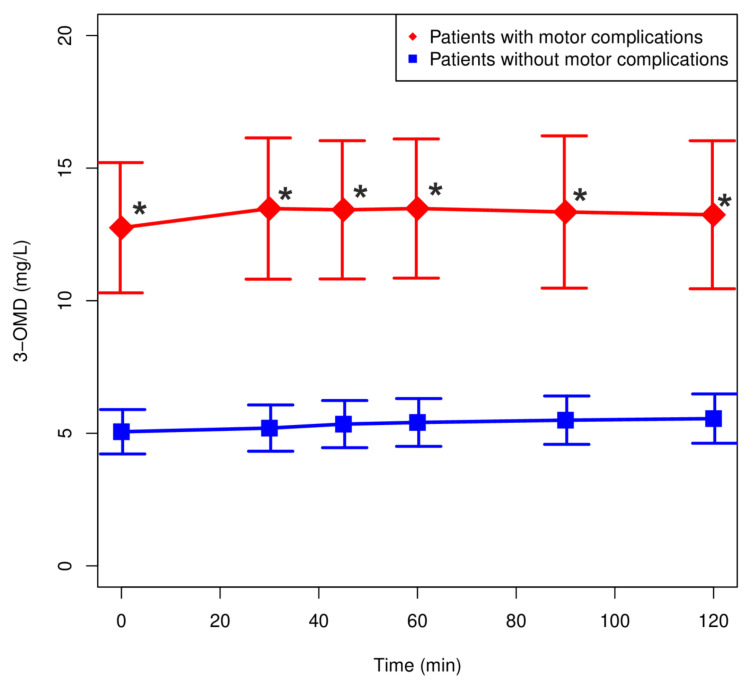
Plasma concentration: time profiles (mean values ± SEM) for 3-OMD of patients with motor complications (fluctuations and/or dyskinesias) and without motor complications. * *p* < 0.05.

**Table 1 pharmaceutics-13-01395-t001:** Demographic data of patients administrating both levodopa and ropinirole, and patients administrating levodopa but not ropinirole.

Demographic and Clinical Data (Mean, SD, Range)	Patients Administrating Both Levodopa and Ropinirole (*n* = 13)	Patients Administrating Levodopa But Not Ropinirole (*n* = 14)	*p*-Value
Sex (M/F)	10 M/3 F	10 M/4 F	Ns
Age (yr)	68.4 ± 7.3 57–83	70.9 ± 10.8 45–86	Ns
Weight (kg)	79.3 ± 16.4 60–113	79.8 ± 21.6 50–110	Ns
UPDRS score	25.3 ± 9.9 10–44	21.8 ± 10.6 6–36	Ns
Disease duration (yr)	8.5 ± 5.9 1–21	5.3 ± 3.2 1–11	Ns
Levodopa treatment duration (yr)	7.2 ± 4.9 0.5–20	3.9 ± 3.5 0.5–11	Ns
Current levodopa dosage (mg/d)	692.9 ± 245.6 400–1100	592.9 ± 200.8 300–950	Ns
Ropinirole treatment duration (yr)	3.3 ± 2.5 1–8	-	-
Current ropinirole dosage (mg/d)	7.3 ± 2.2 4–10	-	-
Motor complications (fluctuations and/or dyskinesias)	8 Yes/5 No	4 Yes/10 No	Ns

*p*-values calculated by means of Mann–Whitney or Fisher exact test, where appropriate; Ns, not significant. Interval variables are presented as mean ± SD followed by range.

**Table 2 pharmaceutics-13-01395-t002:** Demographic data of patients with motor complications (fluctuations and/or dyskinesias) and without motor complications.

Demographic and Clinical Data (Mean, SD, Range)	Patients with Motor Complications (Fluctuations or Dyskinesias) (*n* = 12)	Patients without Motor Complications (Fluctuations and/or Dyskinesias) (*n* = 15)	*p*-Value
Sex (M/F)	8 M/5 F	13 M/2 F	Ns
Age (yr)	72.6 ± 9.0 57–86	67.1 ± 8.8 45–79	Ns
Weight (kg)	68.2 ± 12.3 50–85	88.7 ± 18.6 51–113	<0.002
UPDRS score	28.1 ± 9.2 15–44	20.2 ± 9.8 6–35	Ns
Disease duration (yr)	10.5 ± 4.7 6–21	3.8 ± 2.4 1–8	<0.001
Levodopa treatment duration (yr)	9.1 ± 4.0 4–20	2.4 ± 1.9 0.5–6	<0.001
Current levodopa dosage (mg/d)	811.5 ± 167.3 600–1100	496.7 ± 158.6 300–900	<0.001
Ropinirole treatment duration (yr)	3.6 ± 2.6 1–8 (*n* = 8)	2.0 ± 1.4 1–3 (*n* = 5)	Ns
Current ropinirole dosage (mg/d)	8.0 ± 2.2 4–10 (*n* = 8)	6.0 ± 1.4 4–8 (*n* = 5)	Ns

*p*-values calculated by means of Mann–Whitney or Fisher exact test, where appropriate; Ns, not significant. Interval variables are presented as mean ± SD followed by range.

**Table 3 pharmaceutics-13-01395-t003:** Results of the Wald test of significance of the covariates.

PK Parameter	Covariate	Significance (*p*)
$T_{lag}$	Weight	0.013
	Movement disorder	<0.001
$k_{10}$	Age	0.007
$k_{20}$	Ropinirole therapy	0.019
V/F	Sex	<0.001
	Ropinirole concentration at 2 h	<0.001

**Table 4 pharmaceutics-13-01395-t004:** Means of pharmacokinetic parameters of levodopa and 3-OMD after administration levodopa and ropinirole (*n* = 13) and levodopa but not ropinirole (*n* = 14).

Parameter	Patients Administrating Both Levodopa and Ropinirole (*n* = 13)	Patients Administrating Levodopa but Not Ropinirole (*n* = 14)	*p*-Value
	**levodopa**	**levodopa**	
lag time (min)	9.97 ± 11.3	16.6 ± 10,6	Ns
*V/F* (L)	32.4 ± 10.3	36.1 ± 11.8	Ns
*CL/F* (L/min)	0.816 ± 0.323	0.835 ± 0.336	Ns
*AUC*_0–∞_ (mg×min/L)	150 ± 83	147 ± 83	Ns
*t*_max_ (min)	35.5 ± 34.6	46.4 ± 26.8	Ns
*C*_max_ (mg/L)	2.27 ± 1.56	1.79 ± 0.95	Ns
*MRT* (min)	41.6 ± 6.6	45.1 ± 6.9	Ns
*k*_a_ (min^−1^)	0.18 ± 0.16	0.31 ± 0.42	Ns
*k*_10_ (min^−1^)	0.0213 ± 0.0038	0.0195 ± 0.0034	Ns
*k*_12_ (min^−1^)	0.00333 ± 0.0056	0.00318 ± 0.00040	Ns
*k*_20_ (min^−1^)	0.000262 ± 0.000043	0.000204 ± 0.000029	0.002
	**3-OMD**	**3-OMD**	
*t*_max 3-OMD_ (min)	305 ± 207	304 ± 152	Ns
*MET*_3-OMD_ (min)	3997 ± 666	5040 ± 612	0.001
*C*_max 3-OMD_ (mg/L)	0.476 ± 0.316	0.423 ± 0.188	Ns
*AUC*_3-OMD [0–__∞]_ (mg×min/L)	1993 ± 1277	2232 ± 1032	Ns

*p*-values calculated by means of ANOVA on log-transformed data; Ns—not significant.

**Table 5 pharmaceutics-13-01395-t005:** Pharmacokinetic parameters of levodopa and 3-OMD of patients with motor complications (fluctuations and/or dyskinesias) and without motor complications.

Parameter	Patients with Motor Complications (Fluctuations and/or Dyskinesias) (*n* = 12)	Patients without Motor Complications (Fluctuations and/or Dyskinesias) (*n* = 15)	*p*-Value
	**levodopa**	**levodopa**	
lag time (min)	1.51 ± 0.19	22.9 ± 4.1	<0.0001
*V/F* (L)	28.2 ± 10.9	39.2 ± 8.7	0.01
*CL/F* (L/min)	0.674 ± 0.330	0.948 ± 0.271	0.01
*AUC*_0–∞_ (mg×min/L)	191 ± 103	114 ± 34	0.01
*t*_max_ (min)	30.5 ± 31.2	49.6 ± 28.5	0.02
*C*_max_ (mg/L)	2.47 ± 1.56	1.66 ± 0.90	Ns
*MRT* (min)	44.8 ± 8.1	42.2 ± 5.7	Ns
*k*_a_ (min^−1^)	0.19 ± 0.18	0.29 ± 0.40	Ns
*k*_10_ (min^−1^)	0.0197 ± 0.0042	0.0208 ± 0.0032	Ns
*k*_12_ (min^−1^)	0.00327 ± 0.00047	0.00324 ± 0.00050	Ns
*k*_20_ (min^−1^)	0.000247 ± 0.000046	0.000220 ± 0.000044	Ns
	**3-OMD**	**3-OMD**	
*t*_max 3-OMD_ (min)	290 ± 174	315 ± 261	Ns
*MET*_3-OMD_ (min)	4260 ± 812	4759 ± 784	Ns
*C*_max 3-OMD_ (mg/L)	0.580 ± 0.324	0.343 ± 0.101	0.01
*AUC*_3-OMD [0–__∞]_ (mg×min/L)	2588 ± 1425	1740 ± 688	0.04

*p*-values calculated by means ANOVA on log-transformed data; Ns—not significant.

## Data Availability

The data that support the findings of this study, except for patients’ identifiers, are available from the corresponding author upon reasonable request.

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
