# Peer review of "Pharmacokinetics of Levodopa and 3-O-Methyldopa in Parkinsonian Patients Treated with Levodopa and Ropinirole and in Patients with Motor Complications"

_pharmaceutics, 2021, doi:10.3390/pharmaceutics13091395_

Round 1

Reviewer 1 Report

The authors conducted a pharmacokinetic study of levodopa (L-dopa) and its major metabolite, 3-O-methyldopa (3-OMD), in patients with Parkinson's disease. The authors aimed to investigate the possible interaction between L-dopa and ropinirole, as well as, to correlate the pharmacokinetic parameters of L-dopa and 3-OMD with the occurrence of motor complications. According to the authors, the plasma concentrations of L-dopa and 3-OMD were not affected by ropinirole, but when comparing such concentrations between the patients with or without motor complications, the 3-OMD plasma concentrations of patients presenting motor dysfunctions were significantly higher. The manuscript is well-written, the introduction provided the basis to understand the rationale behind the work. The abstract is quite long. Discussion should be improved. The conclusion section is missing

Below are some specific comments:

Introduction

  • Page 2 line 61: According to the authors: “The inhibition of AADC only allowed 5-10% an oral dose of LD to reach the central nervous system [1,6].” Such a claim cannot be justified by reference 6, which deals with the effect of amino acids on the uptake of LD, therefore, it should be deleted. Moreover, the original statement from reference 1 is “Despite being pivotal in LD metabolism, the inhibition of AADC only allowed 5-10% of an oral dose of LD to reach the central nervous system [28, 59].” I suggest, therefore, that authors review the statement, and the original references are 28 and 59 mentioned by Tambasco et al. Maybe authors should rephrase: “Even though inhibition of AADC is a clinical strategy during LD therapy, it has been demonstrated that only 5-10% of an oral dose of LD reach the central nervous system when combined with an AADC inhibitor (REFERENCES).”
  • Reference 4: Please replace “…pharmacodynamik…” to “…pharmacodynamic…”

  1. Materials and Methods

2.1. Patients

  • Page 3 line 95: Please, include the mean body weight, separate the weight according to the treatment as was done for age (Table 1). The difference in CL, VD, Cmax, AUC may be correlated with the difference of mean body weight and age since elderly people have a lower renal function! The author should discuss this issue under the discussion section. Check the reference for comparison: 1016/0002-9343(89)90023-5. In this study, the differences in Cmax and AUC of ciprofloxacin was correlated with the difference of the body weight, and in the CL was correlated with the age

3.2. Plasma concentration of levodopa and 3-OMD

1) Page 7, Figure 2, and 3 and Figures S2 and S3: I would suggest using log scale on Y-axis. Additionally, include the error bars and figures 2 and 3.

2) Please change the pattern of the connecting lines of the figures from Smooth lines to Straight lines.

3.3. Pharmacokinetic population modelling

  • Page 9: I would suggest including the micro constant values Ka, K10, and K12 in table 4. Is it possible to present and compare such micro constant between the patients with and without motor dysfunctions?
  • Please define the time interval used to calculate the AUC on tables 4 and 5. For example AUC0-∞ ?

  1. Discussion

1) The first and second paragraphs of the discussion section are repeating the information already mentioned in the introduction and material and methods, please be more objective, and be focused on discussing the results which, for instance, are very interesting.

2) Pages 10-11 lines 305-312: According to the authors: “Concentrations of levodopa and its metabolite 3-OMD were higher in patients treated with levodopa and ropinirole, but the differences were not statistically significant. There was no significant effect of ropinirole on the concentration of levodopa or its metabolite. With the exception of the mean exit time of 3-OMD there were no significant differences in the pharmacokinetic parameters in patients treated with levodopa and ropinirole compared to patients who did not receive ropinirole. Taylor et al. [37] showed no pharmacokinetic interaction between ropinirole and levodopa. We observed significantly shorter mean-residence-time (MET3-OMD) for 3-O-methyldopa in patients treated with levodopa and ropinirole.” Some questions raised: i) The patients under L-dopa therapy but no ropinirole, were also taking inhibitors of AADC (benserazide and carbidopa), since the L-dopa and 3-OMD plasma concentration of patients under ropinirole therapy was not significantly different from those patients that received L-dopa plus the AADC inhibitors, if possible that ropinirole could also inhibit the AADC? The authors should discuss this possibility. In the study conducted by Taylor et al, patients did not receive AADC inhibitors, moreover, according to Taylor et al, “The L-dopa Cmax increased on average by 16% when given with ropinirole. The 95% CI of 0.92, 1.47 was wide, indicating that an average increase in Cmax of up to 47% is possible.” ii) Authors should discuss possible reasons that lead to the reduction of MET3-OMD in patients treated with L-dopa and ropinirole compared to that of patients that received L-dopa.

3) Page 11, line 335: According to the authors: “As reported by other authors, our results showed that 3-OMD, which is a major metabolite of levodopa, plays a role in inducing movement disorders during long-term levodopa therapy” I would suggest rephrasing the sentence: “As reported by other authors (include the references), our results suggest that 3-OMD, which is a major metabolite of levodopa, plays a role in inducing movement disorders during long-term levodopa therapy”.

Author Response

Response to Reviewer 1 Comments

Abstract has been shortened, discussion has been corrected, conclusion  added

Introduction

  • Introduction was changed according to suggestions,  suggested literature position added.
  • Reference 4 has been corrected.

Materials and Methods

2.1. Patients

Please, include the mean body weight, separate the weight according to the treatment as was done for age (Table 1)

Body weight is now included in Table 1 as well as in Table 2 in 2.1 Patients section.

3.2. Plasma concentration of levodopa and 3-OMD

1) Page 7, Figure 2, and 3 and Figures S2 and S3: I would suggest using log scale on Y-axis. Additionally, include the error bars and figures 2 and 3.

Log scale is inappropriate for l-dopa, since there are several zero concentrations (i.e. below LOQ). We don’t think that mixing linear and log scales in the presentation is a good choice, so 3-OMD is also plotted in linear scale. Error bars were added; they represent the standard error of the mean (SEM) as error bars with SD might worsen the plot readability.

2) Please change the pattern of the connecting lines of the figures from Smooth lines to Straight lines.

The figures were redrawn according to the Reviewer’s suggestions.

3.3. Pharmacokinetic population modelling

Page 9: I would suggest including the micro constant values Ka, K10, and K12 in table 4.

All microconstants are now included in Tables 4 and 5

 Is it possible to present and compare such micro constant between the patients with and without motor dysfunctions?

Done.

  1. Discussion

Discussion was corrected as suggested by the Reviewer 1.

Conclusion added.

Reviewer 2 Report

This manuscript discusses the pharmacokinetics of levodopa and 3-O-methyldopa in Parkinsonian patients treated with levodopa and ropinirole and in patients with motor complications. 

The manuscript contains a few grammatical/language errors that have been indicated in the PDF copy of the manuscript with highlighted text and notes. 

The authors also need to address the following: 

  1. Line 121:  the authors indicate that a dose of 100 mg of levodopa was administered to the patients.  Why was this dose used?  This should be explained as the average dose that the patients normally received as chronic therapy was much higher than 100 mg.  Furthermore, the half-life of 3-OMD is approximately 15 hours, therefore the presence of 3-OMD could primarily be attributed to these patients’ chronic treatment rather than the 100 mg dose administered as part of the study.  This observation is based on the fact that a 12 hour period of medication withdrawal was allowed in the study.  This places a question on the validity of the conclusions that the authors drew with regards to 3-OMD and the symptoms of the patients.  The authors should clearly indicate that the symptoms attributed to 3-OMD is in all likelihood related to the chronic doses and not attributed to the dose used in the study unless they can prove otherwise. 
  2. The authors should clearly indicate the novelty of their research in the abstract and the conclusions of this manuscript.

Author Response

Response to Reviewer 2 Comments

The manuscript contains a few grammatical/language errors that have been indicated in the PDF copy of the manuscript with highlighted text and notes.

English language and style has been improved. The indicated errors were corrected as suggested. Thank you!

  1. Line 121:  the authors indicate that a dose of 100 mg of levodopa was administered to the patients.  Why was this dose used?  This should be explained as the average dose that the patients normally received as chronic therapy was much higher than 100 mg.

The daily dose is typically much higher, indeed, but it results as a sum of multiple doses, which are typically of order, or equal to small multiple of 100 mg, what is also a typical dose of the pharmaceutical product.

Furthermore, the half-life of 3-OMD is approximately 15 hours, therefore the presence of 3-OMD could primarily be attributed to these patients’ chronic treatment rather than the 100 mg dose administered as part of the study.  This observation is based on the fact that a 12 hour period of medication withdrawal was allowed in the study.  This places a question on the validity of the conclusions that the authors drew with regards to 3-OMD and the symptoms of the patients.

The background of previous administrations was carefully included in our calculations. However, after each administration (including administration of the test dose) small changes in concentration of 3-OMD are observed and they provide information on model parameters.

  1. Abstract has been shortened and improved, conclusion has been added, as suggested.

Round 2

Reviewer 2 Report

I am satisfied with the changes done.  

Author Response

Manuscript:
- minor language/style improvements

References:
- corrected numeration
- spacing corrections
- removed doubled spaces
